# The Impact of Urodynamic Findings on Fatigue and Depression in People with Multiple Sclerosis

**DOI:** 10.3390/biomedicines13030601

**Published:** 2025-03-01

**Authors:** Anke K. Jaekel, Julius Watzek, Jörn Nielsen, Anna-Lena Butscher, John Bitter, Marthe von Danwitz, Pirmin I. Zöhrer, Franziska Knappe, Ruth Kirschner-Hermanns, Stephanie C. Knüpfer

**Affiliations:** 1Department of Neuro-Urology, Clinic for Urology, University Hospital Bonn, 53127 Bonn, Germany; 2Department of Neuro-Urology, Johanniter Neurological Rehabilitation Center Godeshoehe GmbH, 53177 Bonn, Germany; 3Department of Cognitive Rehabilitation, Neurological Rehabilitation Center Godeshoehe GmbH, 53177 Bonn, Germany; 4Department of Medical Psychology | Neuropsychology and Gender Studies, Center for Neuropsychological Diagnostics and Intervention (CeNDI), Faculty of Medicine and University Hospital Cologne, University of Cologne, 50937 Cologne, Germany; 5Clinic St. Hedwig, Department of Paediatric Urology, University Medical Center Regensburg, 93053 Regensburg, Germany

**Keywords:** multiple sclerosis, neuro-urology, bladder, urodynamics, fatigue, depression

## Abstract

**Background**: Fatigue and depression are common symptoms of multiple sclerosis (MS) that severely impair quality of life. The factors influencing both are of increasing interest for establishing therapeutic synergisms. Correlations between the symptoms of neurogenic lower urinary tract dysfunction (NLUTD), fatigue, and depression have been described, but the impact of pathological urodynamic study (UDS) findings has not been investigated to date. **Method**: This retrospective, observational study correlated UDS findings of 274 people with MS (PwMS), prospectively collected between February 2017 and September 2021, with scores on the Fatigue Scale for Motor and Cognitive Functions and the German version of the Centre for Epidemiologic Studies Depression Scale. The effects of abnormal UDS on the FSMC and ADS scores were examined. Abnormal UDS was defined as follows: first desire to void (FDV) < 100 mL, strong desire to void < 250 mL (SDV), abnormal sensation, detrusor overactivity, detrusor–sphincter dyssynergia, reduced cystometric bladder capacity < 250 mL (MCBC), and compliance < 20 mL/cm H_2_O (C_low_). **Results**: PwMS with C_low_ (mean difference 3.21, 95% CI 0.25; 6.17, *p* = 0.036) or FDV < 100 mL (mean difference 2.61, 95% CI 0.1; 5.12, *p* = 0.041) had significantly higher FSMC mean values than those without. PwMS with MCBC < 250 mL (relative risk 1.06, 95% CI 1.02; 1.1, *p* = 0.006) or C_low_ (relative risk 1.06, 95% CI 1.02; 1.1, *p* = 0.004) had an increased risk of clinically relevant fatigue. No effects were found for depression. **Conclusions**: PwMS with NLUTD have higher FSMC scores and an increased risk of fatigue in our retrospective study. The assessment of prospective longitudinal data regarding the effect of successfully treated NLUTD on fatigue is important for utilising therapeutic synergisms for improved quality of life in PwMS.

## 1. Introduction

Fatigue in people with (Pw) multiple sclerosis (MS) is one of the most common and distressing symptoms of the disease [1]. This subjective lack of physical and mental energy is reported to have a prevalence of up to over 90% in PwMS [1]. Fatigue is difficult to treat [2], leads to a loss of health-related quality of life, and has a major impact on the earning capacity of those affected [1]. MS-related fatigue can be categorised as physical and cognitive fatigue. Physical fatigue is based on muscle weakness, which leads to physical exhaustion [3]. Cognitive fatigue, which can occur even before the diagnosis of MS, is characterised by a decline in cognitive activities due to poor concentration, memory problems, or emotional instability [3].

Another coexisting neuropsychiatric symptom of MS is depression, which affects almost one in two PwMS [4]. The common causes are thought to be MS-related inflammatory effects with demyelination, axonal damage, and brain atrophy [4]. Secondary causes are sleep disorders, medication side effects, and physical and psychological stress due to neurogenic symptoms of the bowel or lower urinary tract [5].

Neurogenic lower urinary tract dysfunction (NLUTD) may be present at first diagnosis [6] or may develop with disease progression in up to 80% of PwMS [6]. The localisation and severity of the inflammatory lesions in the central nervous system can cause various impairments of storage and voiding function. Detrusor overactivity (DO) is the most common, with a frequency of 43–62% [7]. Detrusor–sphincter dyssynergia (DSD) occurs in 25–35% of cases, and detrusor underactivity (DU) or detrusor acontractility is present in around 20% [7,8]. These underlying pathologies can result in different lower urinary tract symptoms (LUTS). In addition to musculoskeletal symptoms, impaired visual acuity, and concentration, as well as impaired bowel and sexual function, these LUTS lead to a high level of physical and psychosocial stress [9,10] with a significant reduction in quality of life [11].

Various studies have investigated the relationship between neurogenic LUTS and fatigue [5,9,10,12]. Significant correlations were also found between LUTS and depression in PwMS [13]. The design of these studies was based on the recording of neurogenic LUTS from medical history, questionnaires, or bladder diaries. These data can be influenced by individual factors of the persons affected. The reported severity and level of resulting distress can also be affected by the individual’s current personal situation [14]. In addition, LUTS described by those affected do not allow any reliable conclusions to be drawn about the objective underlying type of NLUTD [8,15]. No studies have been published so far regarding objective NLUTD and fatigue. Therefore, we aimed to investigate the relationships between urodynamic findings and fatigue and depression in detail to achieve a better understanding of the interactions among these MS symptoms.

## 2. Patients and Methods

### 2.1. Patients

This study included 274 PwMS who were treated in the neuro-urological department of an inpatient neurological rehabilitation centre between February 2017 and September 2021 and met the inclusion criteria. The urological data were prospectively collected by the Department of Neuro-Urology, and the data regarding fatigue and depression were collected and provided by the Department of Cognitive Rehabilitation. These data were retrospectively analysed for this study.

The inclusion criteria were as follows: age of at least 18 years, definitive diagnosis of MS according to the McDonald criteria [16], the mental ability to answer questions, and provision of written informed consent. Inclusion in the study was independent of the clinical course or severity of the disease. The exclusion criteria were as follows: persons under the age of majority, pregnant or breastfeeding women, persons with untreated acute lower urinary tract infections, and persons who refused to provide written informed consent.

This study was conducted in accordance with the Declaration of Helsinki. All patients provided written informed consent. Ethical approval (EK 313/13-University Hospital Bonn) was obtained.

### 2.2. Fatigue Assessment

We assessed the presence and severity of fatigue using the Fatigue Scale for Motor and Cognitive Functions (FSMC) [3]. This scale focuses on motor and cognitive fatigue and contains 20 items (10 for each subscale) with a Likert scale from never (1) to almost always (5). Fatigue of clinical relevance is defined as a total score of ≥43 or ≥22 in the motor or cognitive subgroups [3].

### 2.3. Depression Assessment

Depression was assessed using the Centre for Epidemiologic Studies Depression Scale [17] (CES-D). In this instance the German version, Allgemeine Depressionsskala (ADS), was used [18]. This scale is a 20-item questionnaire for self-assessment of depressive symptoms that refers to the previous 2 weeks [18]. Each item is answered using a four-point Likert scale (rarely (0) to mostly (3)), and a total score of >22 indicates clinically relevant depression in the normal German population. The ADS focuses on cognitive items rather than somatic and fatigue-related items, which helps to avoid issues with measurement due to an overlap of depression and fatigue symptoms.

### 2.4. Urodynamic Studies

Urodynamic studies (UDSs) were performed according to ICS standards [19,20] using an MMS Nexam Pro Urodynamic System (Laborie/Medical Measurement Systems B.V., Enschede, The Netherlands). All examinations were performed by a small, highly experienced team of nurses and physicians specialised in neuro-urology and urodynamic studies. The UDS were always performed using the same clinical procedure according to the current standards of the International Continence Society [19,20].

We defined the UDS findings indicative of NLUTD according to the current doctrine [19,20] and according to previous studies [21] as follows:First desire to void < 100 mL;Strong desire to void < 250 mL;Any abnormal sensation;Maximum cystometric bladder capacity < 250 mL;Bladder compliance < 20 mL/cm H_2_O;Presence of any type of DO or DSD.

Detrusor hypocontractility/underactivity is not included in the definition of UDS findings indicative of NLUTD because the diagnosis cannot be made based on a urodynamic measurement alone.

In addition to a detailed descriptive analysis of the cohort and the urodynamic findings, we analysed the effects of the urodynamic findings on fatigue and depression in the following ways:The effects of urodynamic findings indicative of NLUTD on clinically relevant fatigue/depression (FSMC and ADS scores over the cut-off values mentioned above) and on the continuous scores for fatigue (FSMC sum score, FSMC motor and cognitive subscale) and depression (ADS);Effects of single urodynamic parameters (indicative of NLUTD) on clinically relevant fatigue/depression (FSMC and ADS scores over the cut-off values as mentioned above) and on the continuous scores for fatigue (FSMC sum score, FSMC motor and cognitive subscales) and depression (ADS).

### 2.5. Statistical Analysis

To investigate the effects of abnormal UDS findings on clinically relevant fatigue and depression, we used logistic regression models and determined the relative risk between people with clinically relevant fatigue/depression with and without abnormal UDS findings. The influence of abnormal UDS findings on the continuous scores of FSMC and ADS was determined by comparing the corresponding mean values between the two groups using t-tests and linear models adjusted for gender and age. The effect of the qualities (normal, increased, reduced) of abnormal sensation on clinically relevant fatigue and depression was analysed using logistic models. A likelihood ratio test was first used to investigate whether the individual abnormal sensitivity had any significant explanatory power for the presence of depression/fatigue; only if this test was significant were the individual qualities of sensitivity compared with each other.

All the analyses were performed using the R statistical programming language [22]. Results with *p*-values < 0.05 were considered statistically significant.

## 3. Results

### 3.1. Patients and Their Disease Characteristics

The cohort consisted of 66.8% (183/274) female and 33.2% (91/274) male patients. The mean Expanded Disability Status Scale (EDSS) score was 4.25 (SD 1.59; min 1, max 8). We had no information on the EDSS in 19.3% (53/274) of our cohort. The mean age was 48.31 years (SD 10.36; min 18, max 73). The mean duration of the disease at the time of data collection was 10.82 years (SD 8.92; min 0, max 47), while 2.6% (7/274) of the patients did not report their disease duration. The course of MS was primary progressive in 10.2% (28/274), relapsing–remitting in 55.8% (153/274), and secondary progressive in 31.4% (86/274). Additionally, 2.6% (7/274) did not specify their course of MS.

No UDS data were provided by 9.1% (25/274) of the patients. Table 1 provides a detailed overview of the UDS findings.

Furthermore, 84.8% (207/244) had abnormal UDS findings indicative of NLUTD according to the predefined definition; 15.2% (37/244) did not. The individual urodynamic parameters indicative of NLUTD were distributed as follows: FDV < 100 mL 23.8% (58/244), SDV < 250 mL 34.8% (78/224), MCBC < 250 mL 13.3% (33/249), compliance < 20 mL/cm H_2_O 5.0% (12/240), DSD 36% (81/225), and DO 32.8% (82/250). Abnormal sensation was found in 72.0% (180/250) (hyposensitive 24.0% 60/250; hypersensitive 47.2% 118/250; asensitive 0.8% 2/250). Normosensitivity was found in 28.0% (70/250); we had no sensation data in 8.8% (24/274) of cases.

A high percentage of the study population suffered from clinically significant fatigue based on the FSMC and the subscales. Approximately half of the PwMS showed clinically relevant depression. A detailed analysis is shown in Table 2.

The frequency distribution of the individual UDS parameters in PwMS with and without fatigue or depression is shown in Figure 1.

### 3.2. Effects of UDS Findings on Fatigue and Depression

The analysis showed an increased risk of clinically significant fatigue in all three FSMC subscales for PwMS with UDS findings indicative of NLUTD, but we could not confirm this statistically. This effect was not found for clinically significant depression (Table 3). The mean values for the individual continuous FSMC scores (total, cognitive, motor) or ADS showed no statistically significant difference between PwMS with and without UDS findings indicative of NLUTD (Table 3). Thus, we could not demonstrate that PwMS with abnormal UDS had higher scores for fatigue or depression in our cohort.

### 3.3. Effects of the Individual UDS Parameters Indicative of NLUTD on FSMC and ADS

Among the individual UDS parameters, PwMS with a compliance < 20 mL/cm H_2_O or with FDV < 100 mL showed significantly higher mean values in FSMC_mot_ compared to PwMS with a higher compliance or an FDV ≥ 100mL. Table 4 provides an overview of the correlations between the individual UDS parameters indicative of NLUTD and the continuous scores for FSMC and ADS.

### 3.4. Effects of Individual Urodynamic Parameters Indicative of NLUTD on Fatigue and Depression

The assessment of the effect of the individual UDS parameters indicative of NLUTD on the presence of clinically relevant fatigue and depression showed the following results: MCBC < 250 mL and compliance < 20 mL/cm H_2_O showed an increased relative risk for fatigue in the FSMC sum score and in the motor subscale. Compliance < 20 mL/cm H_2_O also showed this effect in the FSMC cognitive subscale. Our data did not show an increased relative risk for the other UDS parameters. Again, no effect was shown for depression. An overview of the results is given in Table 5.

Finally, we analysed the effects of abnormal sensation on clinically relevant fatigue and depression. We had no data on fatigue or depression for the two asensory PwMS, so these were not included in the analysis. For the other qualities of abnormal sensation (normal, hypo-, and hypersensitive), we found significant differences in the pairwise comparison of risk of clinically significant fatigue for all FSMC categories, but not for depression. In almost all cases (except for FSMC _mot_), individuals with normal sensation showed the lowest risk of fatigue compared to the other qualities. PwMS with an increased sensation showed the highest risk. Figure 2 provides an overview of the risks; the detailed results of the analysis can be found in Appendix A.

## 4. Discussion

Improving the disease-related quality of life of PwMS has become more important in recent years [5,10,11]. In line with this, correlations between neurogenic dysfunction of the bowel and urinary tract and neuropsychological symptoms have already been identified [9,10,13]. However, these studies have always been based on patient-reported symptoms. To our knowledge, the correlation between fatigue and UDS findings, which represent the manifestation of NLUTD, has not been investigated to date. Furthermore, the relationship between depression and detrusor overactivity in urodynamics was reported as a secondary finding in a study on sexual dysfunction associated with MS [23].

For the present study, we therefore assessed the UDS findings of 274 PwMS and correlated them with scores for fatigue and depression. The urodynamic results were used as influencing variables and fatigue and depression as target variables. In our cohort, people with UDS findings indicative of NLUTD showed higher risks for clinically relevant fatigue, but this effect could not be statistically verified in our sample. For depression, we were unable to show any effects of the UDS findings indicative of NLUTD on the continuous or binary ADS scores.

Various studies have described significant correlations between LUTS, recorded in questionnaires, and depression in PwMS [13,24]. In contrast, only a few correlations between depression and LUTS were found when documented more objectively using a bladder diary [12]. Here, nocturia and urgency showed a statistically significant effect on depression. The other parameters analysed, such as the voided volume or incontinence, did not show a correlation. Nocturia can be explained by sleep disturbances in depression. This relationship between depression and nocturia has already been described [25]. Urgency, on the other hand, is the parameter in the bladder diary that is most subjective [12]. A study assessing the correlation between sexual dysfunction and urodynamic results in PwMS examined depression in relation to the occurrence of DO [23]. Like our study, that study also found no differences in depression depending on the presence or absence of DO [23].

Regardless of our findings, the additional burden of urgency symptoms and frequent toilet visits may contribute to depression. This bidirectionality of LUTS and depression has been shown several times in people who do not suffer from MS [26,27]. The treatment of LUTS is described as having a positive effect an depression [28]. A lack of physical activity can also have a negative impact on depression [29], and there is evidence that LUTS restricts the mobility of PwMS [30]. Considering the existing literature and our results, we conclude that depression itself influences the assessment of the severity and the perception of LUTS. It seems to be less an underlying shared physical cause for both conditions. However, further studies are needed to clarify the directionality and causality of these complex inter-relationships.

In our analysis, PwMS with UDS findings indicative of NLUTD showed an increased risk of clinically relevant fatigue, but we were unable to provide statistical evidence based on our data. This may be because the majority of the people included had UDS findings indicative of NLUTD. Almost 95% of the PwMS in our cohort had fatigue. This means that the reference group without these characteristics was very small. Furthermore, it must be noted that fatigue is anything but a clearly defined disorder and that questionnaires greatly overestimate the phenomenon [31]. It contains a large amount of other information, e.g., aspects of secondary fatigue. However, this high percentage of patients affected by fatigue is reflected in various studies [1,32]. It would therefore be useful to include the urodynamic data of outpatients with MS in further studies, expecting a lower share suffering from fatigue in this population. Furthermore, longitudinal observation of the fatigue scores in the same person with and without NLUTD therapy could help in obtaining clearer results. Perhaps other assessment tools for fatigue or depression should be chosen. The FSMC, due to the choice of items, is more likely to measure the trait characteristics (permanently occurring in everyday life) of fatigue; the ADS, as a dimensional screening questionnaire, only measures depressive symptoms from the last week and is rather a depressive ‘mood barometer’.

In our cohort, the appearance of reduced bladder compliance and first desire to void were associated with higher fatigue scores. PwMS with reduced bladder compliance, reduced MCBC, or hypersensitivity showed an increased risk of clinically relevant fatigue in our statistics. Common to all these individual urodynamic parameters is the impairment of the bladder’s storage function [33,34]. Analogous results were shown in a previous study, which correlated fatigue with LUTS based on bladder diaries [12]. The symptoms urgency and reduced voided volume had the most correlations. Other authors focussing on urgency symptoms in PwMS demonstrated the negative influence on psychosocial and cognitive aspects [13]. In their work, Carotenuto et al. found a strong association between cognitive impairment in the Symbol Digit Modality Test (SDMT), the Neurogenic Bowel Dysfunction Score, and the Actionable Bladder Symptom Screening Tool. This association justified the authors to initiate investigations of bladder and bowel function in PwMS who show an abnormal SDMT [35].

In our work, the UDS parameter that correlates the most with fatigue is compliance <20 mL/cm H_2_O. Based on currently available data, it is not clear why low compliance has these effects on fatigue. Studies on UDS and the correlation with psychosocial and cognitive disorders are rare [36]. No results on the correlations between individual urodynamic parameters have been published yet. In contrast to DO or DSD, reduced compliance represents a serious consequential damage to bladder elasticity due to various disorders of the storage and emptying function of the bladder [34]. Therefore, reduced compliance can be subject to a wide variety of influences. Common associations of MS-specific characteristics with fatigue or reduced compliance do not provide a satisfactory explanation either. For example, there are contradictory statements as to whether fatigue is dependent on the duration or course of MS [37,38]. There is no consistent information in the literature on the relationship between MS characteristics and the severity of NLUTD [15,39,40]. We did not find correlations between reduced compliance and patient- and MS-specific characteristics in our study cohort.

Overall, this study was the first to systematically investigate the influence of UDS findings indicative of NLUTD on fatigue and depression in PwMS. We were able to show some correlations for fatigue but not for depression. Issues regarding causality cannot be answered due to the study design. These must be the subject of further and specifically targeted research.

## 5. Limitations

There are several limitations of our study that should be addressed. The main restriction of our study is the high share of people suffering from fatigue (nearly 95%). This meant that few subjects without fatigue were available for statistical comparison. This limited the correlation analysis as one of the analysed characteristics was present in the majority of cases.

Furthermore, there was selection bias, as each of the included PwMS was participating in inpatient neurological rehabilitation at the time of the study. This could have led to the inclusion of particularly severe cases of PwMS who needed rehabilitation. This emphasises the need for a longitudinal study of the relationship between fatigue/depression and NLUTD treatment in the same individual in order to minimise the influence of the aforementioned limitations.

Another limitation is the lack of detrusor hypocontractility/underactivity in the definition of UDS findings indicative of NLUTD. As the ongoing subject of various studies, hypocontractility/underactivity is still difficult to define in UDS. Our definition of hypocontractility/underactivity is based on the assessment of the examining physician including information from bladder diaries and uroflowmetry. Therefore, this parameter was not included in further analysis in the current study.

Moreover, we did not examine the influence of gender on the effects of NLUTD on fatigue and depression. However, gender-specific differences are an interesting scientific topic and would require a separate extensive analysis. Finally, the recording and evaluation of fatigue represents a limitation because all instruments for capturing fatigue are based on the self-reports of those affected and are therefore subjective [31].

## 6. Conclusions and Further Directions

This study found significant correlations between fatigue and low compliance, reduced cystometric bladder capacity, and hypersensitivity. In contrast, no effects of urodynamic findings indicative of NLUTD on depression were identified, although correlations between both symptoms have been described before. Due to the increasing relevance of therapeutic synergisms and quality of life in PwMS, it is important to assess the interactions and directionality of individual MS symptoms in more detail. Therefore, prospective studies are important to investigate the effects of successfully treated NLUTD on distressing symptoms such as fatigue in a longitudinal approach. Based on our results, this assessment of NLUTD should include urodynamic findings in addition to questionnaires and bladder diaries. In order to clarify the causality of the relationship between fatigue, depression, and NLUTD, the correlation of neural damage visualised by neutral imaging modalities with urological and neuro-psychiatric symptoms is required. This may enable a more specific understanding of the multidirectional relationships between symptoms and identify new therapeutic approaches.

## Figures and Tables

**Figure 1 biomedicines-13-00601-f001:**
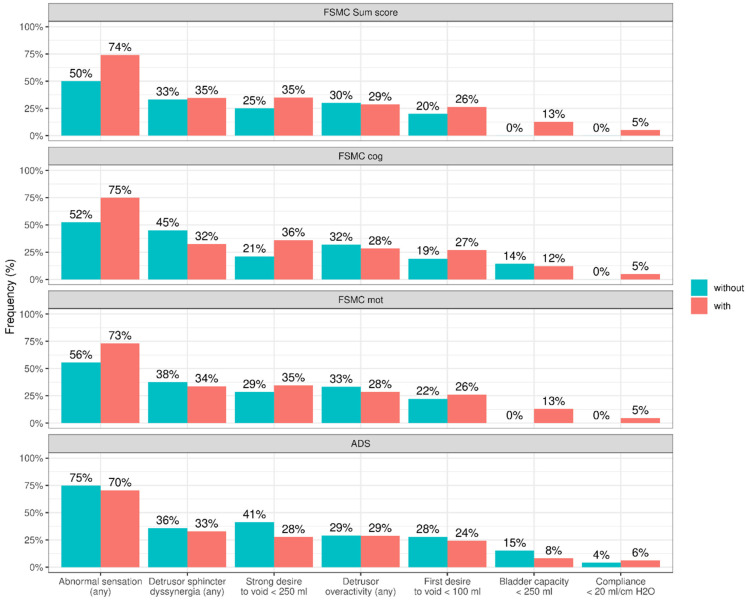
Frequency distribution of the individual urodynamic parameters indicative of NLUTD in PwMS in the groups with/without fatigue or depression.

**Figure 2 biomedicines-13-00601-f002:**
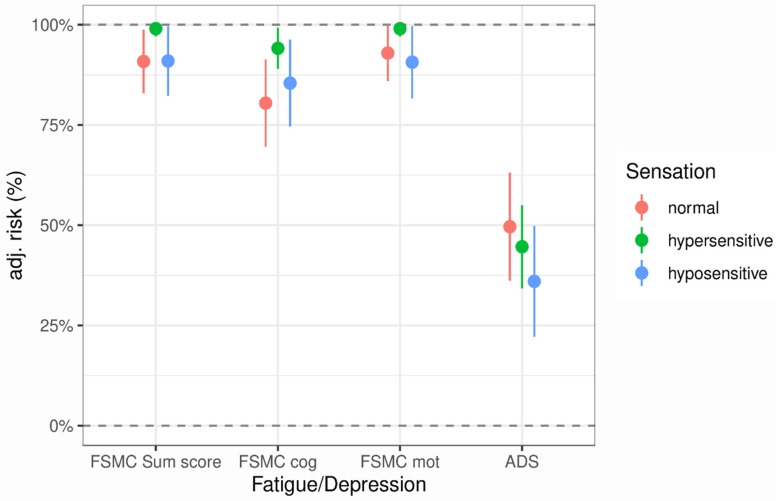
Graphical presentation of the fatigue/depression risks according to different qualities of sensation.

**Table 1 biomedicines-13-00601-t001:** Descriptive analysis of the UDS findings of the study cohort.

	Mean (SD)	Median (25–75%)	Min–Max	Valid Values % (N)	Missing % (N)
FDV [mL]	195.2 (121.5)	169 (104; 263.5)	8; 710	89.4 (245)	10.6 (29)
SDV [mL]	292.7 (115.7)	291.5 (2.5; 5)	47; 588	81.8 (224)	18.2 (50)
Compl [mL/cmH_2_O]	76.2 (66.7)	61.4 (2.0; 14.5)	4; 453	86.9 (238)	13.1 (36)
MCBC [mL]	401.9 (142.0)	406.5 (311.3; 485)	41; 1000	90.5 (248)	9.5 (26)
Reflex volume [cm H_2_O]	198.2 (109.5)	200 (109.8; 268.8)	10; 449	29.9 (82)	70.1 (192)
Q_max_ [mL/s]	13.2 (6.6)	13 (8; 18)	2; 37	46 (126)	54 (148)
MV [mL]	330.4 (159.9)	323 (214; 439.5)	9; 715	83.6 (229)	16.4 (45)
	**% (N)**	**Valid values % (N)**	**Missing % (N)**		
DSD	29.6 (81)	82.1 (225)	17.9 (49)		
DO	29.9 (82)	91.2 (250)	8.8 (24)		

FDV, first desire to void; SDV, strong desire to void; Compl, bladder compliance; MCBC, maximum cystometric bladder capacity; Qmax, maximum uroflow; MV, micturition volume; DSD, detrusor–sphincter dyssynergia; DO, detrusor overactivity. N, number of PwMS

**Table 2 biomedicines-13-00601-t002:** Description of fatigue and depression scores in our study population.

	Missing % (N)	Mean (SD)	Median (25–75%)	Min–Max	Clinically Significant % of Valid Observations (N) *
ADS	23.7% (65)	22.49 (11.44)	21 (14; 31)	1; 54	47.4 (99)
FSMC_sum score_	22.3% (61)	75.53 (16.84)	79 (66; 89)	22; 99	94.8 (202)
FSMC_cog_	23.7% (65)	36.56 (9.8)	39 (31; 44)	10; 50	95.2 (199)
FSMC_mot_	23.7% (65)	38.94 (8.18)	41 (36; 45)	11; 50	88.5 (185)

* Clinically significant fatigue is defined with a sum score of ≥43 or ≥ 22 in the motor or cognitive subgroups. A total score of >22 indicates clinically relevant depression in the normal German population. N, number of PwMS

**Table 3 biomedicines-13-00601-t003:** Correlation of UDS findings indicative of NLUTD with FSMC/ADS scores (continuous and binary).

	FSMC_sum score_ ≥ 43 *	FSMC_cog_ ≥ 22 *	FSMC ≥ 22_mot_ *	ADS > 22 *
	Relative Risk (Yes vs. No) (95% CI) *p*-Value			
UDS findings indicative of NLUTD	1.12 (0.96; 1.29) *p* = 0.145	1.12 (0.92; 1.36) *p* = 0.253	1.07 (0.95; 1.22) *p* = 0.279	0.99 (0.61; 1.58) *p* = 0.951
	**FSMC_sum score_**	**FSMC_cog_**	**FSMC_mot_**	**ADS**
	***t*-test mean diff (95% CI) *p*-value**			
UDS findings indicative of NLUTD	1.81 (−6.17; 9.79) *p* = 0.648	0.84 (−3.82; 5.5) *p* = 0.718	0.93 (−2.79; 4.65) *p* = 0.616	0.45 (−4.66; 5.57) *p* = 0.858

* Clinically significant fatigue is defined with a sum score of ≥43 or ≥22 in the motor or cognitive subgroups. A total score of >22 indicates clinically relevant depression in the normal German population.

**Table 4 biomedicines-13-00601-t004:** Correlation of individual UDS parameters indicative of NLUTD with continuous FSMC and ADS scores.

UDS Findings	FSMC_sum_ _score_			FSMC_cog_			FSMC_mot_			ADS		
	Mean (SD)		*t*-Test (95% CI) *p*-Value	Mean (SD)		*t*-Test (95% CI) *p*-Value	Mean (SD)		*t*-Test (95% CI) *p*-Value	Mean (SD)		*t*-Test (95% CI) *p*-Value
	with	without		with	without		with	without		with	without	
FDV < 100 mL	79.06 (15.4)	74.38 (17.46)	4.68 (−0.59; 9.95) *p* = 0.081	38.12 (8.96)	36.07 (10.18)	2.06 (−1.04; 5.15) *p* = 0.19	40.9 (7.14)	38.28 (8.56)	2.61 (0.1; 5.12) *p* = 0.041	23.83 (12.13)	22.3 (11.38)	1.53 (−2.46; 5.52) *p* = 0.447
SDV < 250 mL	77.2 (15.27)	75.79 (16.99)	1.41 (−3.65; 6.46) *p* = 0.582	37.07 (9.04)	36.68 (10.24)	0.39 (−2.64; 3.43) *p* = 0.798	40.07 (7.14)	38.93 (8.13)	1.14 (−1.26; 3.54) *p* = 0.349	22.22 (12.35)	23.66 (11.13)	−1.44 (−5.28; 2.4) *p* = 0.459
Abn. Sens.	76.5 (15.99)	73.46 (19.28)	3.04 (−2.93; 9.01) *p* = 0.313	37.08 (9.38)	35.63 (11.03)	1.45 (−1.99; 4.89) *p* = 0.405	39.39 (7.84)	37.83 (9.21)	1.56 (−1.31; 4.43) *p* = 0.282	22.78 (11.38)	22.47 (12.07)	0.31 (−3.57; 4.19) *p* = 0.875
MCBC < 250 mL	77.17 (15.07)	75.46 (17.09)	1.71 (−5.25; 8.67) *p* = 0.62	36.39 (10.17)	36.72 (9.87)	−0.33 (−4.96; 4.29) *p* = 0.884	40.78 (5.78)	38.69 (8.34)	2.09 (−0.69; 4.87) *p* = 0.136	19.77 (11.37)	23.07 (11.58)	−3.3 (−8.61; 2.01) *p* = 0.213
Compliance < 20 mL/cm H_2_O	82.11 (9.14)	75.09 (17.19)	7.02 (−0.26; 14.31) *p* = 0.057	40.25 (6.67)	36.35 (10)	3.9 (−1.76; 9.55) *p* = 0.152	42.0 (3.38)	38.79 (8.29)	3.21 (0.25; 6.17) *p* = 0.036	23.89 (9.71)	22.36 (11.4)	1.53 (−6.03; 9.09) *p* = 0.659
DO	74.53 (17.57)	75.85 (16.82)	−1.33 (−6.83; 4.18) *p* = 0.634	35.74 (10.01)	36.9 (9.87)	−1.15 (−4.34; 2.04) *p* = 0.474	38.5 (8.76)	39.01 (8.06)	−0.51 (−3.26; 2.23) *p* = 0.71	22 (11.81)	22.86 (11.48)	−0.86 (−4.62; 2.89) *p* = 0.648
DSD	74.12 (17.44)	76.78 (16.58)	−2.66 (−8.08; 2.75) *p* = 0.332	35.22 (10.46)	37.48 (9.63)	−2.25 (−5.52; 1.01) *p* = 0.174	38.55 (8.16)	39.4 (8.05)	−0.85 (−3.44; 1.75) *p* = 0.52	22.46 (11.3)	23.09 (11.76)	−0.63 (−4.28; 3.02) *p* = 0.733

FDV, first desire to void; SDV, strong desire to void; Compl, bladder compliance; MCBC, maximum cystometric bladder capacity; DSD, detrusor–sphincter dyssynergia; DO, detrusor overactivity.

**Table 5 biomedicines-13-00601-t005:** The effect of the individual UDS parameters indicative of NLUTD on the presence of clinically relevant fatigue.

UDS Findings	FSMC_sum score_ ≥ 43			FSMC_cog_ ≥ 22			FSMC_mot_ ≥ 22		
	Adjusted Risk (%) (95% CI)		Relative Risk (95% CI) *p*-Value	Adjusted Risk (%) (95% CI)		Relative Risk (95% CI) *p*-Value	Adjusted Risk (%) (95% CI)		Relative Risk (95% CI) *p*-Value
	without	with		without	with		without	with	
FDV < 100 mL	94.2 (90.15; 98.25)	96.3 (90.93; 100.00)	1.02 (0.96; 1.09) *p* = 0.522	86.84 (80.94; 92.75)	90.78 (81.89; 99.67)	1.05 (0.93; 1.17) *p* = 0.456	94.89 (91.00; 98.78)	96.20 (90.57; 100.00)	1.01 (0.95; 1.08) *p* = 0.69
SDV < 250 mL	94.64 (90.30; 98.98)	96.71 (92.09; 100.00)	1.02 (0.96; 1.09) *p* = 0.509	85.59 (78.73; 92.46)	92.39 (85.14; 99.64)	1.08 (0.97; 1.2) *p* = 0.173	95.48 (91.39; 99.57)	96.63 (91.82; 100.00)	1.01 (0.95; 1.08) *p* = 0.707
Abnormal sensation	90.63 (82.64; 98.62)	96.46 (93.25; 99.67)	1.06 (0.97; 1.17) *p* = 0.187	80.37 (69.50; 91.24)	91.21 (86.08; 96.34)	1.13 (0.98; 1.31) *p* = 0.088	92.67 (85.51; 99.83)	96.35 (92.97; 99.74)	1.04 (0.96; 1.13) *p* = 0.353
MCBC < 250 mL	94.47 (90.62; 98.33)	100.00 (100.00; 100.00)	1.06 (1.02; 1.1) *p* = 0.006	87.62 (82.13; 93.11)	86.04 (71.50; 100.00)	0.98 (0.82; 1.17) *p* = 0.841	95.07 (91.25; 98.88)	100.00 (100.00; 100.00)	1.05 (1.01; 1.09) *p* = 0.013
Compliance < 20 mL/cmH_2_O	94.25 (90.46; 98.04)	100.00 (99.99; 100.00)	1.06 (1.02; 1.1) *p* = 0.004	85.89 (80.23; 91.55)	100.00 (99.99; 100.00)	1.16 (1.09; 1.24) *p* < 0.001	94.85 (91.16; 98.54)	100.00 (99.99; 100.00)	1.05 (1.01; 1.1) *p* = 0.008
DO	94.81 (90.78; 98.84)	94.76 (88.89; 100.00)	1.00 (0.93; 1.08) *p* = 0.99	87.48 (81.33; 93.62)	86.78 (77.71; 95.84)	0.99 (0.88; 1.12) *p* = 0.9	95.51 (91.66; 99.35)	94.76 (88.82; 100.00)	0.99 (0.92; 1.07) *p* = 0.83
DSD	94.38 (89.86; 98.9)	95.33 (89.74; 100.0)	1.01 (0.94; 1.09) *p* = 0.788	89.15 (83.02; 95.28)	83.02 (72.61; 93.44)	0.93 (0.81; 1.07) *p* = 0.322	95.24 (91.00; 99.49)	95.21 (89.37; 100.00)	1.00 (0.93; 1.07) *p* = 0.993

FDV, first desire to void; SDV, strong desire to void; Compl, bladder compliance; MCBC, maximum cystometric bladder capacity; DSD, detrusor–sphincter dyssynergia; DO, detrusor overactivity.

## Data Availability

The data presented in this study are available on request from the corresponding author. The data are not publicly available due to privacy.

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
