# Peer review of "The Impact of Urodynamic Findings on Fatigue and Depression in People with Multiple Sclerosis"

_biomedicines, 2025, doi:10.3390/biomedicines13030601_

Round 1

Reviewer 1 Report

Comments and Suggestions for Authors

I have reviewed the study entitled’ The Impact of Urodynamic Findings on Fatigue and Depres- 2 sion in People with Multiple Sclerosis’’.  The study seems intriguing and has the capacity to make significant contributions to both the current body of literature and clinical practice. However, the article needs some revisions.

1.    Abstract:  Methods: should include the time interval of the study.

 2.    Abstract:  Methods: should include the type of the study.

 3.    Introduction: While informative, it could benefit from a clearer articulation of the knowledge gap the study aims to fill.

 4.    Introduction: should be shorter and decrease the number of references (there are many references for this section). Write the introduction in three separate paragraphs.

--first, general information                                                                                   

-- second: study-specific sentences + study's hypothesis,                                    

-- third: We aimed to..                                                                                   

5.     Introduction/discussion: Add or replace at least 5 references dated 2023–2025 with relevant existing ones.

 6.    Discussion: There should not be sentences that are exactly like the introduction and conclusion; instead, it should include a study-specific discussion in light of current references (2023–2025).

 7.    Tables: Identify which tests correlate with which p-values.

 8.    Minor grammatical corrections are needed for clarity, e.g., simplifying long sentences in the discussion and methods sections.

Comments on the Quality of English Language

....

Author Response

Dear Reviewer,

we highly appreciate the chance to improve our manuscript in order to have it published in this special issue.

The comments proved to be very helpful and constructive. This careful review made us aware of critical aspects of our article.

Please find a point-by-point response including information on the changes made to the manuscript below. Comments are in italic, responses are in red, changes to the manuscript have been implemented in track changes function.

Comments and Suggestions for Authors

I have reviewed the study entitled’ The Impact of Urodynamic Findings on Fatigue and Depression in People with Multiple Sclerosis’’.  The study seems intriguing and has the capacity to make significant contributions to both the current body of literature and clinical practice. However, the article needs some revisions.

  1. Abstract:  Methods: should include the time interval of the study.
  2. Abstract:  Methods: should include the type of the study.

We added both to the abstract.

  1. Introduction: While informative, it could benefit from a clearer articulation of the knowledge gap the study aims to fill.

We considered this point to be related to the following one.

  1. Introduction: should be shorter and decrease the number of references (there are many references for this section). Write the introduction in three separate paragraphs.

--first, general information                                                                                   

-- second: study-specific sentences + study's hypothesis,                                    

-- third: We aimed to.     

We have tightened up and re-anranged the introduction, removed some sources and replaced others. Some sources are older, but we have left them as primary sources (Penner et al. 2009, Lin Butler et al. 2019) because they cannot be replaced and play a central role in the rest of the paper. The first source is our measurement method for fatigue, the second one of the few existing papers that has the relation of fatigue, bladder and bowel as a primary outcome.                                                                             

  1. Introduction/discussion: Add or replace at least 5 references dated 2023–2025 with relevant existing ones.

See point 6.

  1. Discussion: There should not be sentences that are exactly like the introduction and conclusion; instead, it should include a study-specific discussion in light of current references (2023–2025).

We would like to thank you for the attentive and careful review of our paper. We have reduced repetitions in the paper.

We have reviewed your comment regarding the current literature carefully and have not found any even more current literature that is helpful to refine the discussion. Literature on the relationship between fatigue, depression and objective bladder dysfunction is generally rare, that’s why we worked at this point with our study. The most recent article on the subject that we are aware of this topic is Walawska-Hrycek et al. 2004 Wiad Lek 2024;77(11):2326-2331. doi: 10.36740/WLek/195752, but this is based on questionnaires again, which was not our aim. Our work was based on the effects of objective bladder dysfunction.  We discussed our results in the context of the most recent literature that was relevant and available for the topic and couldn´t find more current literature, which was useful to raise the quality of the paper.

  1. Tables: Identify which tests correlate with which p-values.

We have added this: Now the tests belonging to the p-values are indicated in the corresponding head columns of the tables.

  1. Minor grammatical corrections are needed for clarity, e.g., simplifying long sentences in the discussion and methods sections.

We changed that.

Reviewer 2 Report

Comments and Suggestions for Authors

This paper describes the role of pathological urodynamic study (UDS) on the correlations between the symptoms of neurogenic lower urinary tract dysfunction (NLUTD) and fatigue. In this paper, UDS findings of 274 people with MS (PwMS)  were correlated with scores on the Fatigue Scale for Motor and Cognitive Functions and Depression Scale of the Centre for Epidemiologic Studies. It was concluded that PwMS with NLUTD have higher FSMC scores and an increased risk of fatigue, but there are no effects for depression. It is an interesting topic to investigate multiple sclerosis (MS) by using UDS, but there is one unclear matter as follows:

1.      In Figure 2, it is shown that the risks for fatigue and depression depend on the quantity of FSMC and ADS scores with sensations.

     Why do you have a difference of the risk between the case of FSMC_cog and the case of FSMC_mot?  It is better to discuss it in this paper from the academic point of view. Please explain in detail and clarify it in the draft paper at discussion section .

Author Response

Dear Reviewer,

we highly appreciate the chance to improve our manuscript in order to have it published in this special issue.

The comments proved to be very helpful and constructive. This careful review made us aware of critical aspects of our article.

Please find a point-by-point response including information on the changes made to the manuscript below. Comments are in italic, responses are in red, changes to the manuscript have been implemented in track changes function.

Comments and Suggestions for Authors

This paper describes the role of pathological urodynamic study (UDS) on the correlations between the symptoms of neurogenic lower urinary tract dysfunction (NLUTD) and fatigue. In this paper, UDS findings of 274 people with MS (PwMS) were correlated with scores on the Fatigue Scale for Motor and Cognitive Functions and Depression Scale of the Centre for Epidemiologic Studies. It was concluded that PwMS with NLUTD have higher FSMC scores and an increased risk of fatigue, but there are no effects for depression. It is an interesting topic to investigate multiple sclerosis (MS) by using UDS, but there is one unclear matter as follows:

  1. In Figure 2, it is shown that the risks for fatigue and depression depend on the quantity of FSMC and ADS scores with sensations.

     Why do you have a difference of the risk between the case of FSMC_cog and the case of FSMC_mot?  It is better to discuss it in this paper from the academic point of view. Please explain in detail and clarify it in the draft paper at discussion section.

We have considered your comment long and carefully and have come up with the following procedure.

We discussed hypersensitivity, which appears to be associated with an increased risk of fatigue compared to hypo- and normosensitivity, in the discussion on the topic of ‘Urgency’.

In all subscales of the FSMC it is the same that the increased sensation has a higher risk than the reduced or normal sensation. As the bladder sensation in the UDS is the parameter with the most subjective influence, we do not intend to discuss this in detail, as the paper focuses on the objectifiable parameters of the NLUTD in the form of UDS measurements. For reasons of clarity, we would prefer to discuss this point not in detail, as this brings us into the field of our previous work, in which the topic of ‘urgency’ has already been discussed in detail.

We very much hope that our proposal will meet with your approval.

Reviewer 3 Report

Comments and Suggestions for Authors

Dear Authors, thank you for the opportunity to read your manuscript. It is well written and of great interest. However, I have some comments that I would like to share with you.

Clarify how the measurements were standardized or calibrated. Provide a clearer explanation for the exclusion of certain parameters in the methodology. Discuss further the gender differences in the text. Describe in more detail the clinical significance of your results and future research in this regard.

Comments on the Quality of English Language

Please work on improving the quality of the English language throughout the manuscript, either with the help of a native speaker colleague or with the help of a professional editing agency.

Author Response

Dear Reviewer,

we highly appreciate the chance to improve our manuscript in order to have it published in this special issue.

The comments proved to be very helpful and constructive. This careful review made us aware of critical aspects of our article.

Please find a point-by-point response including information on the changes made to the manuscript below. Comments are in italic, responses are in red, changes to the manuscript have been implemented in track changes function.

Comments and Suggestions for Authors

Dear Authors, thank you for the opportunity to read your manuscript. It is well written and of great interest. However, I have some comments that I would like to share with you.

  • Clarify how the measurements were standardized or calibrated.

The Standards of the International Consultation on Incontinence ICS for conducting urodynamic measurements provide internationally accepted benchmarks for the quality of urodynamic procedures. Our measurements are always carried out according to these standards, which also define the calibration process in detail. A detailed description of the processes would go beyond the limits of an original article. We have added this two references and that the measurements were always carried out according to the same standard.

  • Provide a clearer explanation for the exclusion of certain parameters in the methodology.

We added the following part: “Detrusor hypocontractility/underactivity is not included in the definition of UDS findings indicative of NLUTD, because the diagnosis cannot be made on a urodynamic measurement alone.” A further explanation is given in the limitation section.

  • Discuss further the gender differences in the text. Describe in more detail the clinical significance of your results and future research in this regard.

We have already raised this interesting topic of the gender related differences in the effect of NLUTD on fatigue and depression. However, the question of these relationships would be a major study outcome that was not included in the current question and intention. Not least due to the already extensive analysis of the current topic and for the sake of clarity, it should be the subject of a separate investigation, which we will address in a separate analysis.

We added an appropriate note to the limitation section.

Comments on the Quality of English Language

Please work on improving the quality of the English language throughout the manuscript, either with the help of a native speaker colleague or with the help of a professional editing agency.     ……

We have submitted the article to the MDPI English editing service.

Round 2

Reviewer 1 Report

Comments and Suggestions for Authors

It has confirmed that all requested revisions have been appropriately implemented. Therefore, I believe the manuscript is ready for publication.